# Modern Hydrogen Technologies in the Face of Climate Change—Analysis of Strategy and Development in Polish Conditions

**Renata Włodarczyk *** and **Paulina Kaleja**

Faculty of Infrastructure and Environment, Czestochowa University of Technology, Dabrowskiego 69, 42-200 Czestochowa, Poland; paulina.kaleja535@wp.pl
* Correspondence: renata.wlodarczyk@pcz.pl

**Abstract:** The energy production market based on hydrogen technologies is an innovative solution that will allow the industry to achieve climate neutrality in the future in Poland and in the world. The paper presents the idea of using hydrogen as a modern energy carrier, and devices that, in cooperation with renewable energy sources, produce the so-called green hydrogen and the applicable legal acts that allow for the implementation of the new technology were analyzed. Energy transformation is inevitable, and according to reports on good practices in European Union countries, hydrogen and the hydrogen value chain (production, transport and transmission, storage, use in transport, and energy) have wide potential. Thanks to joint projects and subsidies from the EU, initiatives supporting hydrogen technologies are created, such as hydrogen clusters and hydrogen valleys, and EU and national strategic programs set the main goals. Poland is one of the leaders in hydrogen production both in the world and in Europe. Domestic tycoons from the energy, refining, and chemical industries are involved in the projects. Eight hydrogen valleys that have recently been created in Poland successfully implement the assumptions of the "Polish Hydrogen Strategy until 2030 with a perspective until 2040" and "Energy Policy of Poland until 2040", which are in line with the assumptions of the most important legal acts of the EU, including the European Union's energy and climate policy, the Green Deal, and the Fit for 55 Package. The review of the analysis of the development of hydrogen technologies in Poland shows that Poland does not differ from other European countries. As part of the assumptions of the European Hydrogen Strategy and the trend related to the management of energy surpluses, electrolyzers with a capacity of at least 6 GW will be installed in Poland in 2020–2024. It is also assumed that in the next phase, planned for 2025–2030, hydrogen will be a carrier in the energy system in Poland. Poland, as a member of the EU, is the creator of documents that take into account the assumptions of the European Union Commission and systematically implement the assumed goals. The strategy of activities supporting the development of hydrogen technologies in Poland and the value chain includes very extensive activities related to, among others, obtaining hydrogen, using hydrogen in transport, energy, and industry, developing human resources for the new economy, supporting the activities of hydrogen valley stakeholders, building hydrogen refueling stations, and cooperation among Poland, Slovakia, and the Czech Republic as part of the HydrogenEagle project.

**Keywords:** green hydrogen; hydrogen economy; hydrogen technology; hydrogen valleys; Polish Hydrogen Strategy

## 1. Introduction

Hydrogen is a modern energy carrier. Technologies that allow the use of hydrogen in the short term will be a solution to the problems of the deteriorating quality of the natural environment, the depletion of fossil resources, and the ever-growing demand for electricity. Hydrogen in fuel cells burns into water with high efficiency, which makes this fuel very

attractive. Fuel cells, as elements of hybrid systems, can be a source of electricity and heat directly transferred to residential buildings. Such a microgenerator system becomes an alternative to boilers and internal combustion engines [1,2]. Hybrid systems based on fuel cell technology and electrolyzers can work as systems using renewable energy sources (RES), as well as conventional methods of electricity processing, energy storage and "storage" devices, and complex supervision and control systems. Energy from renewable energy sources has great potential, despite the irregularity and/or periodicity of its occurrence. The use of photovoltaic panel/wind turbine—electrolyzer—fuel cell systems increases the energy conversion efficiency of the system for electricity production.

Thanks to the EU Hydrogen Strategy, on 8 July 2020, the Directorate–General for Energy published the Communication from the Commission to the European Parliament, the Council of the European Economic and Social Committee, and the Committee of the Regions "Hydrogen Strategy for a climate-neutral Europe", the production of green hydrogen has been significantly increased by the increase in demand [3]. The hydrogen strategy of July 2020 announced a series of changes to the energy system, which is responsible for 75% of greenhouse gas emissions in the European Union. The strategy sets out ways to use hydrogen through investment, regulation, market creation, research, and innovation. The European Union wants to gradually increase the production of hydrogen and implement it, among others, in the steel, chemical, and transport industries, as well as in the production and storage of energy. The main goal is to achieve such a ceiling that by 2030, the price of green hydrogen will be equal to the cost of producing gray hydrogen. To achieve this, it is necessary to create electrolyzers and develop wind and solar energy-generating infrastructure. As a result, by 2050, the total amount spent by the EU on the development of hydrogen infrastructure may range from EUR 180 to 470 billion. In the adopted hydrogen strategy, the European Commission (EC) assumes the production of up to 1 million tons of hydrogen from RES by 2024 and up to 10 million tons in 2025–2030—then it is to become an important element of the energy system.

In order to present the activities that are already taking place in the development of hydrogen technologies in Poland and in the world, it was necessary to refer to the legal regulations of the European Union and Poland. These documents set the overarching goals of reducing pollution, searching for new energy carriers, and developing innovations. Figure 1 shows the organization of the manuscript.

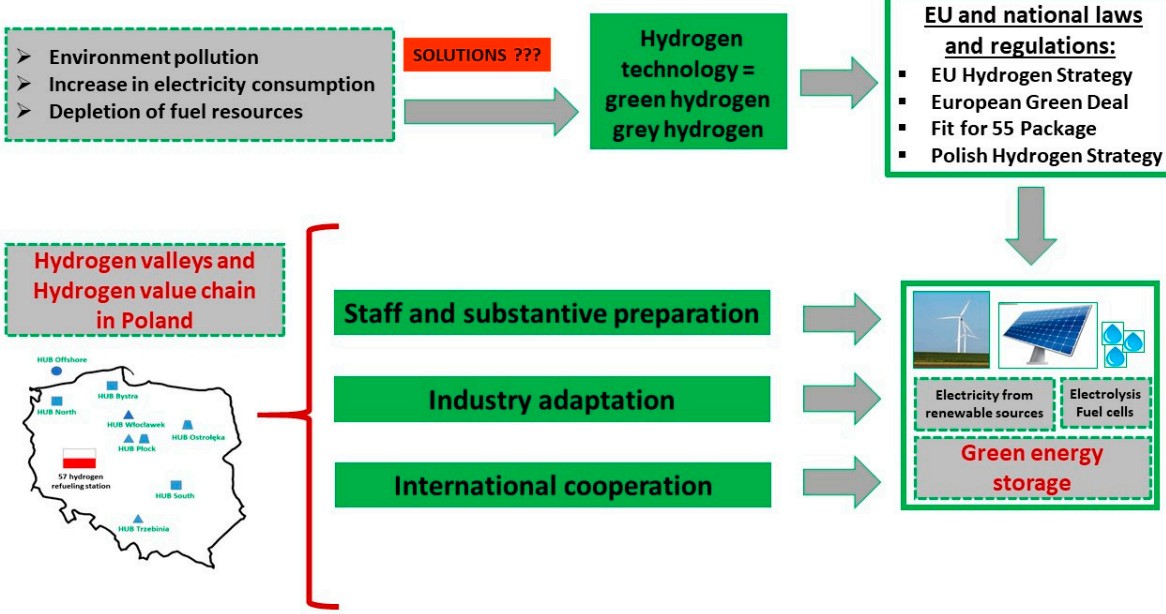

**Figure 1.** Manuscript organization diagram.

### 1.1. The Idea of Using Hydrogen Fuel Cells

There are many innovative solutions for the use of fuel cells in cooperation with generators processing renewable energy sources. It can be noted that these solutions are similar in terms of construction; they differ only in the way they are used [4]. The proposed solutions based on fuel cells are used in such fields as transport, industry, housing, and energy storage [5,6]. As shown in the diagram in Figure 2, combining hydrogen cells with photovoltaic panels into hybrid systems makes it possible to store hydrogen, which is produced in the electrolyzer thanks to the surplus of energy from renewable sources. The cooperation of hydrogen fuel cells with generators processing renewable energy sources (photovoltaic farms, wind farms) is beneficial due to the fact that both technologies complement each other. In the electrolyzer, water decomposes into hydrogen and oxygen under the influence of an electric current. Water for the electrolyzer can be recycled from the fuel cell, where it is a by-product of electrochemical processes. The hydrogen produced in the electrolyzer is either directly sent to the fuel cell or stored [5–8]. In this way, fuel cells can replace photovoltaic panels during high cloud cover or at night. Hydrogen production in the electrolysis process is currently economically and financially unattractive, but economies of scale could reduce the costs of electricity, hydrogen, and necessary equipment generation.

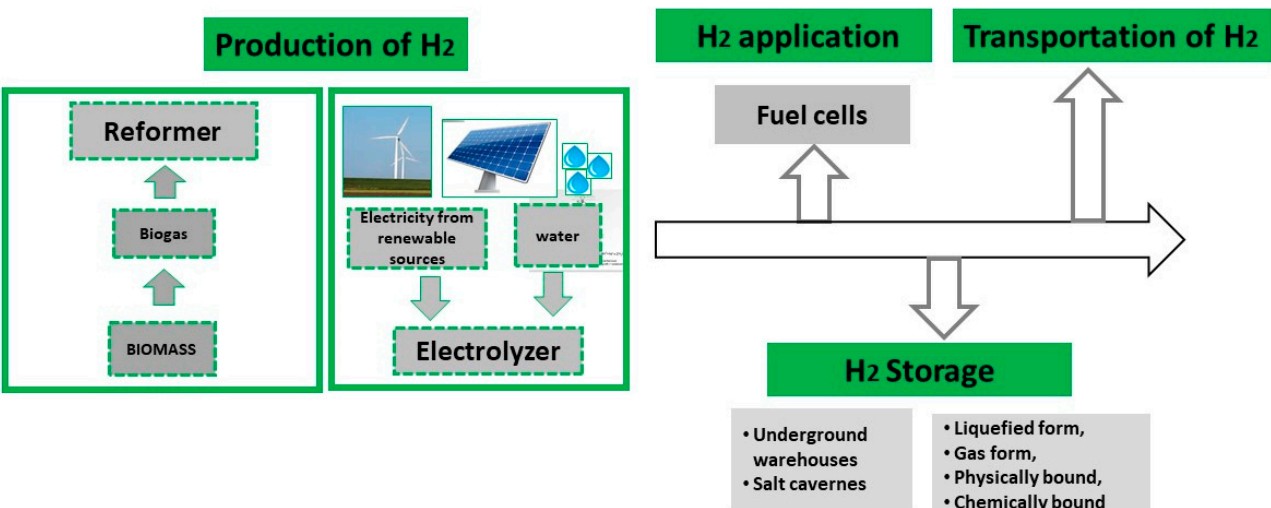

**Figure 2.** The idea of using renewable energy sources for hydrogen technologies [9–11].

### 1.2. RES Electrolyzer Fuel Cells: Green Energy Storage

Fuel cells based on hydrogen fuel can successfully replace combustion generators used in hospitals and generators based on conventional batteries/accumulators. Fuel cells can also be used as a household micro-heat and power plant. The operation of the fuel cell consists of generating electricity resulting from the oxidation reaction of the supplied fuel, which is accompanied by the production of water. In most fuel cells, it uses hydrogen oxidation at the anode and oxygen reduction at the cathode to produce electricity (polymer exchange membrane fuel cells—PEMFC) [12,13].

The generator, based on fuel cell technology, works in a modular system. Its operation is noiseless; only the operation of auxiliary devices can be heard. Thanks to the use of a cell with a solid electrolyte, the system can operate in difficult operating conditions. The cell's operating time is currently 3 years of continuous operation, and by 2025 this time may be extended up to 15 years, with a simultaneous decrease in cell prices [14,15]. Generators that are located near energy consumers also make it possible to use the discharge heat they produce for space heating, water heating, and absorption cooling, which can increase the efficiency of using natural fuels by up to 80%. In the group of low-power fuel cells (from 0.5 to 10 kW), there are about 80 manufacturers who offer pre-commercial and test systems of polymer fuel cells, PEMFC (approx. 85% market share), and oxide, SOFC. These include,

among others, Vaillant GmbH (Germany), EFOY (United States), Hexis AG (Switzerland), UTC Power (United States), MTU CFC Solutions (Germany), and Siemens Westinghouse (United States) [16–18].

Fuel cells combined with an electrolyzer and RES may constitute the so-called green energy warehouse. According to the data from 2021, the installed capacity of the installation for the production of hydrogen in the electrolysis process was estimated at 10 GW, and this amount is still growing due to the installation of further electrolyzers and new projects in EU countries [19,20]. During electrolysis, i.e., the decomposition of water under the influence of an electric current, no pollutants are emitted. Currently, Poland produces about 1 megaton of hydrogen per year, which satisfies about 14% of total consumption in the EU. 50% of hydrogen production is provided by Group Azoty and Lotos. In both companies, hydrogen is used for internal processes: in Group Azoty for the production of ammonia and in Group LOTOS for the needs of a process called hydrocracking. In the context of the demand for hydrogen in Poland and the development of hydrogen-based electromobility, the production of this energy carrier is one of the priorities of the Polish energy sector. The electrolysis process is the most advantageous method of obtaining hydrogen. This process produces a highly pure carrier that is suitable for fuel cells, where the catalysts employed will not be "poisoned" by the use of impure hydrogen. Hydrogen production using an electrolyzer combined with renewable energy sources is already used in many countries where hydrogen strategies have been developed, and the potential of RES allows for the management of energy surpluses. Hydrogen produced from RES is called green hydrogen. European countries are investing in the research and production of hydrogen. Since 2002, the total amount has been PLN 37 billion, and according to forecasts, by 2022 it is expected to reach PLN 600 billion. In Europe, currently only 4% of the hydrogen produced is green hydrogen [21–23].

The development of hydrogen production technology based on electrochemical processes of water molecule decomposition can be divided into individual stages and trends, although the fundamental principles of the technology have remained the same (Figure 3) [24,25]:

- First-generation electrolyzers, used in the years 1800–1950, were used for the production of ammonia; they were alkaline electrolyzers (usually KOH) working at atmospheric pressure; the separator was asbestos; later, the electrolysis process was used for the production of chlorine, where hydrogen became a new by-product;
- Second-generation electrolyzers, used in the years 1950–1980, were based on polymers showing special transport properties—proton exchange. This solution allowed the use of water instead of alkali, which made it possible to reduce the system, reduce its size, and achieve higher efficiency and power density. The leading companies were General Electric, later Hamilton Sundstrand (USA), Siemens, and ABB (Germany);
- Third-generation electrolyzers, used in the years 1980–2010, are characterized by higher efficiency and durability over 50,000 h at lower investment costs and are mostly used for hydrogen production;
- Fourth-generation electrolyzers, developed in the years 2010–2020, are an important element in the process of decarbonization in many industry sectors, not only the energy sector. The constantly increasing capacity of electrolyzer stacks allows for lower capital expenditures (CAPEX), which increases the importance of electrolysis in the energy policy agenda and green hydrogen-specific objectives;
- Fifth-generation electrolyzers, after 2020 to 2050, are most likely to increase electrolyzer capacity from the MW scale to the GW scale, which will be achieved through lower costs (<200 USD/kW) and an increase in lifetime (>50,000 h). These assumptions will require greater production capacity and the rapid, ground-breaking development of research on materials for electrolyzers.

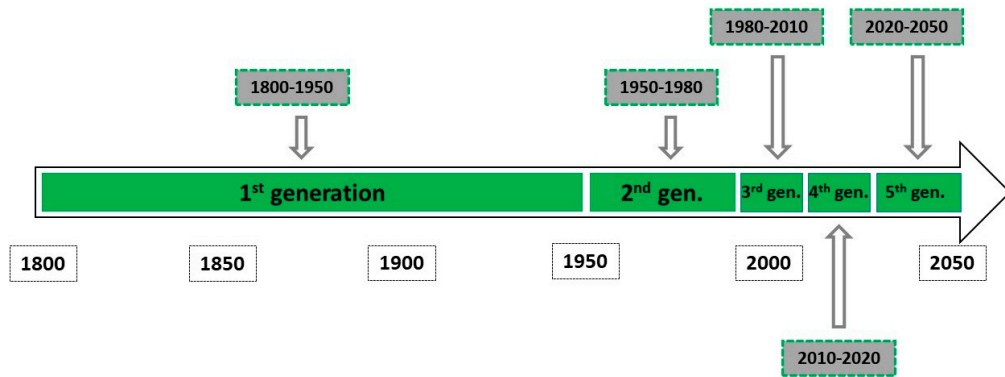

**Figure 3.** Development of technology for electrochemical production of hydrogen in electrolyzers—generations of electrolyzers.

The most important companies producing alkaline electrolyzers include Hymeth, HydrogenPro, and GreenHydrogen. The following companies specialize in the development of PEM electrolyzers: ITM Power, Nel Hydrogen, Siemens, Hydrogenics, and H-Tech. The average amount of hydrogen produced is from about 50.00 to about 90.00 kWh/kg at a pressure of about 35 bar [26]. Solutions based on fuel cell technology are in line with the activities of a number of national and international directives and legal acts, such as the Fit for the F55 package of the European Commission (of 16 July 2021) [27] for the reduction of greenhouse gas emissions, which, in particular, concerns the following assumptions: 50% of hydrogen is to be of renewable origin by 2030, and hydrogen refueling stations are to be placed in the network up to a maximum of 150 km. According to the EC plan, electrolyzers with a capacity of 8.2 GW (57% in Europe) will be built by 2030, and the share of hydrogen in Europe's energy mix will be 13–14% in 2050.

On 8 July 2020, the European Commission announced two documents on energy policy [28]. These documents are in line with the assumptions of the European Green Deal [29]. One of them is the Hydrogen Strategy, and the other concerns the combination of the transport, heating, and electrification sectors and the promotion of clean fuels. The key role in the EU's energy policy is to be played by renewable gases, referred to as "green", including hydrogen. This gas can be used as a fuel, raw material, or energy storage in various branches of the economy, including transport, industry, construction, and energy. During use, hydrogen does not emit pollutants, including carbon dioxide. It is therefore an alternative to the decarbonization of industrial processes. Therefore, its production should be increased, which, as the analyses show, translated into the number of companies interested in hydrogen technologies. According to the International Hydrogen Council, in 2017, the number of companies was 170, and, in mid-2020, it was already 81. Based on the data contained in the report prepared in 2019 by the European Commission [30], it can be observed that the implementation of fuel cell systems depends on the region, including the presence of producers and end-users who need a constant supply of electricity in regions with low grid reliability. The decisive factor affecting the process of implementing fuel cell generators is also the cost that the user will incur for energy. This value depends on the efficiency of the technology, maintenance and operation costs, reliability, and service life.

## 2. Development of the Hydrogen Value Chain Based on Applicable Legal Acts

The decision of the European Union on the need to switch to green ways of obtaining hydrogen contributed to the decision on the need to increase the level of financing for investments, taking into account alternative methods of hydrogen production. The large-scale implementation of hydrogen technologies will undoubtedly bring benefits to the European Union on the international market in the form of the sale of low-emission solutions. Therefore, the legislative work carried out by the European Union has clearly accelerated, and EU countries have started internal measures regulating the cooperation of the hydrogen sector in various sectors of the economy and science. The creation of an

integrated European energy system is associated with the necessity to carry out legislative activities covering many levels. The energy transformation is a long-term undertaking that requires well-thought-out and efficient coordination in all aspects [20]. The European Union formulates laws that are respected in the member states, which indirectly affect the whole world. One of the current challenges faced by the European Parliament is the creation of a legal system that would enable member states to significantly reduce greenhouse gas and pollutant emissions to the environment, based on the principles of just transition and sustainable development. In the process of developing hydrogen technologies, it seems necessary to have a legal road map established by the community of European countries, which will be indispensable in the creation of national legislation. The main goal of the European Green Deal announced on 11 December 2019 is to bring the European Union's greenhouse gas emissions to zero by 2050 and to decouple economic growth from the use of natural resources. The goals of the Paris Agreement and the European Green Deal have been included in the European Climate Law [31,32].

The European Green Deal includes a set of comprehensive legal, economic, and social actions that create a path to the assumed goal. The most important goals set in connection with the energy and economic transformation of the EU are presented in Figure 4. The European Green Deal consists of 10 assumptions [33]:

- Europe without pollution—pollution of air and water—and a solution to the problem of industrial pollution;
- Transitioning to a circular economy—adopting a new circular economy action plan by March 2020;
- "Farm to Fork" program—targets for reducing chemical pesticides (50% by 2030), fertilizers, and increasing the area of organic crops;
- Green Common Agricultural Policy—high environmental and climate ambitions under the reform of the Common Agricultural Policy;
- JUST Transition Mechanism—financial support for regional energy transformation plans;
- Financing transformation—funds for green innovations and public investments;
- Clean, Affordable, and Secure Energy—Assessment of member states' ambitions as part of their national energy and climate plans;
- Achieving climate neutrality—a proposal for the first climate act that records the goal of climate neutrality by 2050;
- Sustainable transport and the adoption of a strategy for sustainable and smart mobility, as well as the review of the Alternative Fuels Infrastructure Directive and the TEN-T Regulation [23];
- Protecting Europe's natural capital—a proposal for an EU biodiversity strategy to 2030.

The climate neutrality of the European Union in such a relatively short time is a huge challenge that requires a long-term strategy, and the fact that time plays a significant role here and, according to the European Council, the transformation of the industry will take 25 years, must be introduced as soon as possible. The participation of the hydrogen sector in the context of the assumptions of the European Council communication on the European Green Deal has great potential. The development of technologies for renewable energy sources is connected with the need to store them during favorable weather conditions and use the stored reserves during a shortage of energy in the power grid. The development of road, rail, and water transport, as well as EU ambitions to reduce the amount of greenhouse gases in this sector, will contribute to greater demand for new hydrogen technologies that can be used as an alternative fuel. Financing research for new technologies for the climate can bring a breakthrough in the context of the use or storage of hydrogen. It should be remembered that all data and predictions regarding the implementation time of individual tasks and phases are valid for the current state of knowledge and possibilities; therefore, they may be significantly shortened or extended due to unexpected events, inventions, or innovations.

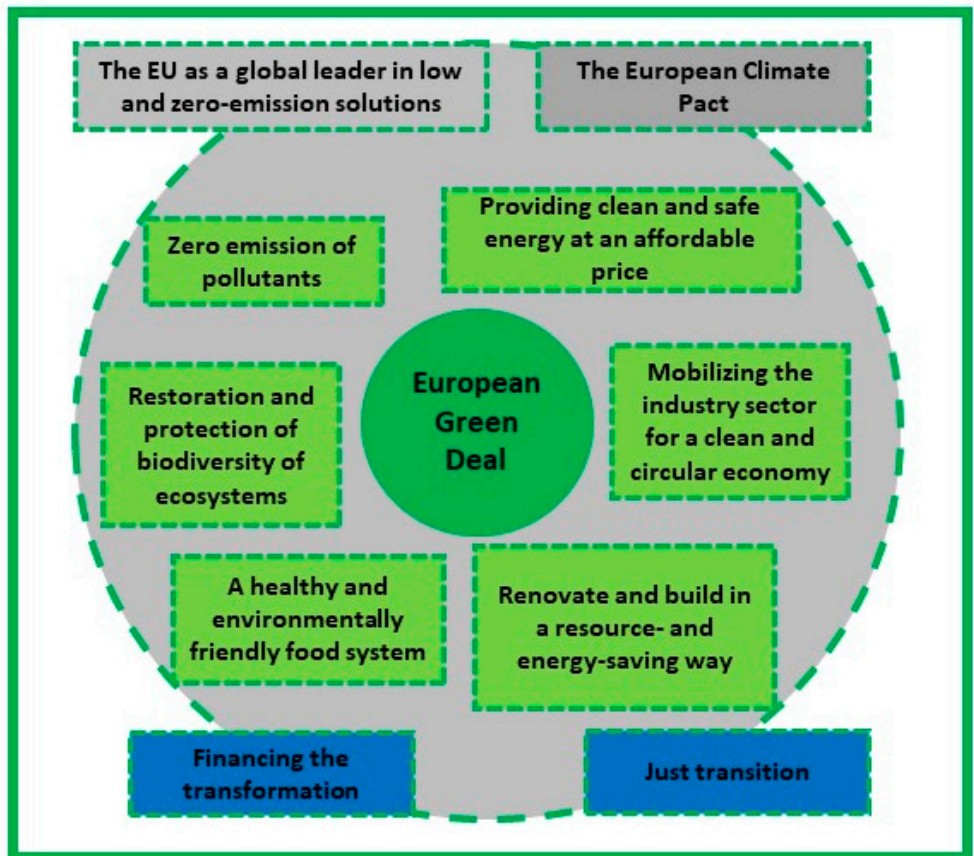

**Figure 4.** Transforming the EU economy for a sustainable future [33].

On 8 July 2020, the European Commission published the "Hydrogen Strategy for a Climate-Neutral Europe". Its main goal is to stimulate the development of the green hydrogen sector so that by 2050 it will be a fully zero-emission and generally available source of energy in the EU [34]. It is a detailed strategy for increasing the share of hydrogen in the European energy mix from 2% to 13–14%. This is a significant increase that will contribute to decarbonization, improve air quality, and reduce $CO_2$ emissions. It raises legal, social, and financial problems. Currently, the low profitability of hydrogen is caused by high production costs and the need to develop the production sector of electrolyzers. However, the great interest in hydrogen is supported by the fact that the number of countries associated with the International Hydrogen Council has increased by 68 countries in two years. One of the benefits of the development of the sector is the large investment outlays that will cause economic growth. The communication distinguishes between the types of hydrogen produced. The division of hydrogen due to the method of obtaining it will be of great importance in the future. Less emission methods will be rewarded more, which will translate into their profitability and, above all, their positive impact on the environment. Gray hydrogen is one of the two types of conventional hydrogen that is produced during coal gasification and can be a by-product of the coking coal production process. The second type is blue hydrogen, which is produced in the process of so-called steam reforming from natural gas. The emissivity of blue and gray hydrogen can be adjusted using carbon dioxide storage and capture or downstream technologies. However, this significantly increases the cost. Green hydrogen is a consequence of electrolysis using electricity from renewable sources, which include wind farms or photovoltaic farms [35].

Due to the scale, scope, and time included in the EU Hydrogen Strategy, individual activities have been included in the following periods-Figure 5 [34]:

- Phase I covers the period from 2020 to 2024, and its goal is to install electrolyzers powered by renewable energy with a capacity of at least 6 GW and a production

capacity of 1 million tons of renewable hydrogen. In this phase, it is planned to increase the production of electrolyzers and build them at demand centers such as steel mills or refineries. A refueling station for buses equipped with fuel cells is to be developed. The years envisaged for this stage will also be the time for creating the hydrogen market, stimulating its liquidity, and planning the benefits of hydrogen investments. The steps taken are centralized and provide for the stationary use of energy to produce hydrogen.

- Phase II covers the period from 2025 to 2030. In these years, the strategic goal is to install electrolyzers powered by renewable energy with a capacity of at least 40 GW and, at the same time, produce 10 million tons of renewable hydrogen in the European Union. Compared to the first phase, which lasted 4 years, this is an almost seven-fold increase, which gives an idea of the dynamics of activities related to the development of the hydrogen economy in this phase. The use of hydrogen will be possible even for more energy-intensive sectors of the economy, such as steel production or heat supply to buildings. At this stage, it is planned to increase price competitiveness. This phase will play an important role in stabilizing the RES-based electricity system by performing supporting and buffering functions.

- Phase III is the period from 2030 to 2050. This is the last phase in which technologies are expected to mature and expand into other sectors of the economy [36–38]. The entire investment plan for the European Union, spread over 30 years, is estimated at EUR 180–470 billion, which consists of the construction of electrolyzers, the construction and connection of new renewable energy sources, refueling stations, and infrastructure. Further funds in the amount of EUR 850–1000 million are to be spent on the creation of 400 hydrogen refueling stations [39,40].

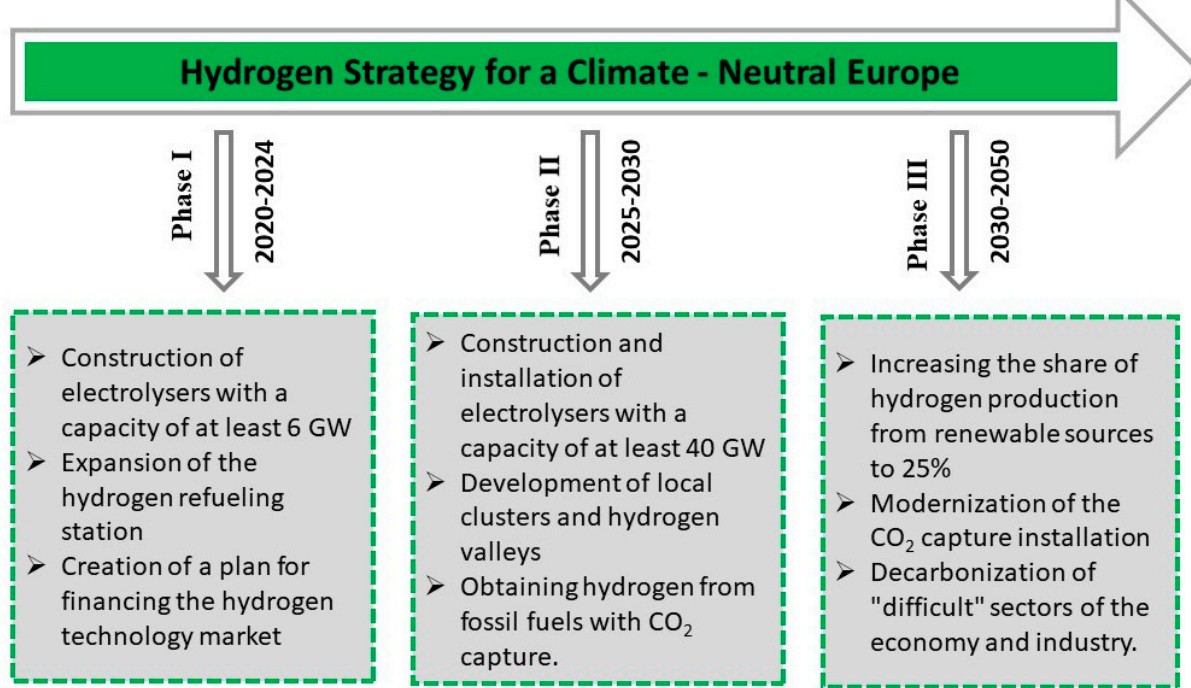

**Figure 5.** Phases of the EU Hydrogen Strategy for 2020–2050.

## 3. Potential and Prospects for the Development of the Hydrogen Market—Good Practices in European Countries

Requirements related to the need to reduce the emission of harmful substances into the environment and prevent their negative effects force politicians to make significant changes in the functioning of domestic and global farms. Surrounded by European countries, these changes are driven primarily by restrictions on the part of the European Union, which

sets directions in the field of fair transformation of energy systems across the continent, leading to the achievement of the goal of a zero-emission economy [41,42]. Directions for activities in this area have been included, among others, in the long-term strategy until 2050 and are based on the use of hydrogen as one of the pillars of the energy transformation. More and more countries are investing in activities related to the research, implementation, and general development of hydrogen technologies, in particular hydrogen production. As a result, by 2050, the total amount spent by the EU on the development of hydrogen infrastructure may range from EUR 180 to 470 billion. The European Union assumes the production of up to 1 million tons of hydrogen from RES by 2024 and up to 10 million tons in 2025–2030, which is to contribute to the inclusion of hydrogen in the general energy system. The European Union wants to gradually increase the production of hydrogen and implement it, among others, in the steel, chemical, and transport industries, as well as in the production and storage of energy. The main goal is to achieve such a ceiling that, by 2030, the prices of green hydrogen will be equal to the costs of production of gray hydrogen (obtained from natural gas). To achieve this, it is necessary to create electrolyzers and develop wind and solar energy-generating infrastructure. In addition, the European Green Hydrogen Acceleration Center (EGHAC) is being implemented [43]. This is an initiative of EIT InnoEnergy that will intensify the production of green hydrogen and help eliminate price differences between gray and green hydrogen. In addition to helping to achieve European carbon neutrality, EGHAC will create up to 500,000 new jobs related to hydrogen infrastructure.

As mentioned earlier, the development of the energy industry and hydrogen technologies in particular is becoming an important goal for individual EU countries. European concerns, including Total, Shell, Edison, BP, and Total, have planned to launch 67 hydrogen production projects in 2020–2030, 62 of which are to use electrolysis thanks to the use of energy from RES [44]. Transmission operators from Italy, Sweden, Spain, Denmark, France, Belgium, the Czech Republic, and Germany want to support the development of the industry. These assumptions will help implement the EU climate policy by enabling the use of surplus energy from wind farms and photovoltaic farms to produce hydrogen. Germany, which consumes the most natural gas among the countries in the community, sees hydrogenation as an opportunity for decarbonization, creating new jobs and benefits for the industry. The German hydrogen strategy of 10 June 2020, prioritizes green hydrogen [45]. Green hydrogen was recognized as the only solution meeting the needs of sustainable development. The government's goal is to achieve 5 GW of electrolysis capacity by 2030 and 10 GW by 2040. This will translate into the production of 14 TWh of hydrogen per year. The strategy envisages supporting the replacement of fossil fuels with hydrogen in the metallurgical and chemical industries as well as in heavy transport. The Netherlands, where gas production is declining, wants to use the potential of the hydrogen industry [46]. The reason is the long-term exploitation of gas, which led to the formation of empty chambers. These chambers have changed the structure of the ground so much that the Groningen region is experiencing cyclical, recurring earthquakes. Regarding the attributes remaining after gas extraction, i.e., experience, infrastructure, and industrial efficiency, the Netherlands intends to use them for the production of (green) hydrogen. The Netherlands wants to produce around 75,000 tons of green hydrogen by 2025. The electricity used to produce hydrogen is to have a capacity of 500 MW, and 5 years later, 3–4 GW—the NorthH2 project. Ultimately, 300,000 cars are to be powered by transport. By 2025, 50 hydrogen refueling stations are to be built; currently, there are three [47]. Spain plans to install 4 GW of electrolyzers by 2030, which is one-tenth of the EU target. The local energy giant, Iberdrola, is building the largest project of this type in Europe [48]. The first model of hydrogen production in Italy assumes the production of hydrogen at the place of its use in order to minimize transport costs, assuming the production of electricity from RES and then hydrogen from electrolysis [49]. The second model assumes the production of hydrogen on site and the transport of electricity. Electricity generated in the south of Italy by wind turbines is then transported through the electricity network to the point of

consumption. The third model assumes centralized production with hydrogen transport. Italian energy companies ENI and ENEL have launched pilot projects for the production of green hydrogen. Two electrolyzers with a capacity of 10 MW each are to be built.

Table 1 presents the assumptions for hydrogen production in selected EU countries in 2025–2030 and expenditures in EUR [42–49]. On the basis of the data contained in the table concerning the power of electrolyzers, 5th generation electrolyzers will be used, which will produce an average of over 75,000 tons of green hydrogen.

**Table 1.** Assumptions for hydrogen production in selected EU countries in 2025–2030 and outlays in Euro [42–49].

| Country | Production/ton | Electrolysis Capacity/GW | Planned Expenditures in Billion Euro |
|---|---|---|---|
| Netherlands | 75 thousand | 3–4 | 0.5 |
| Germany | 10 million | 7–8 | 9 |
| Sweden | 5 million | 4–6 | thousand |
| Spain | 500 thousand | 4 | 8.9 |
| France | No data | 6.5 | 2 |
| Portugal | 170 thousand | 1 | thousand |
| Poland | 165 thousand | 1.7 | 3.7 |

Initiatives to include green hydrogen in energy and industry are covered by programs that are financed by governments and co-financed with EU resources. Figure 6 shows a map of hydrogen projects for 2022. In 2023, the data could change significantly, as hydrogen technologies at each stage of the hydrogen value chain are still being developed and transformed. As part of the France Relance plan for a hydrogen strategy, France plans to allocate approximately EUR 7.2 billion by 2022 [50]. The use of hydrogen as an energy carrier will allow the implementation of three priorities: de-carbonization of industry, the use of hydrogen as a fuel in public transport, and financing research work on the hydrogen industry. Denmark intends to use wind farms to produce hydrogen [51]. Orsted, Siemens Gamesa, and ITM Power want to produce green hydrogen from offshore wind farms as part of the Oyster project, which received financial support from the European Commission fund in the amount of EUR 5 million. The funds will be used to finance electrolyzers to be used as part of a project to assess the potential for green hydrogen production from offshore wind farms. In the future, their work will be integrated with the technology of desalination of sea water in order to obtain it for hydrogen production. An interesting solution is international cooperation between the countries of southern Europe, where the use of solar energy dominates, and the countries around the North Sea basin, where there is a high potential for energy production from wind energy. A similar cooperation was established under the name of the project HydrogenEagle between Poland, Slovakia, and the Czech Republic. The cooperation is based on the production of green hydrogen using biomass and photovoltaic farms in electrolyzers with a capacity of 1 GW until 2030 [52].

The hydrogen sector is at an early stage of development, but it has great potential, and the multitude of applications in various industries allows us to believe that further development will be beneficial in many respects.

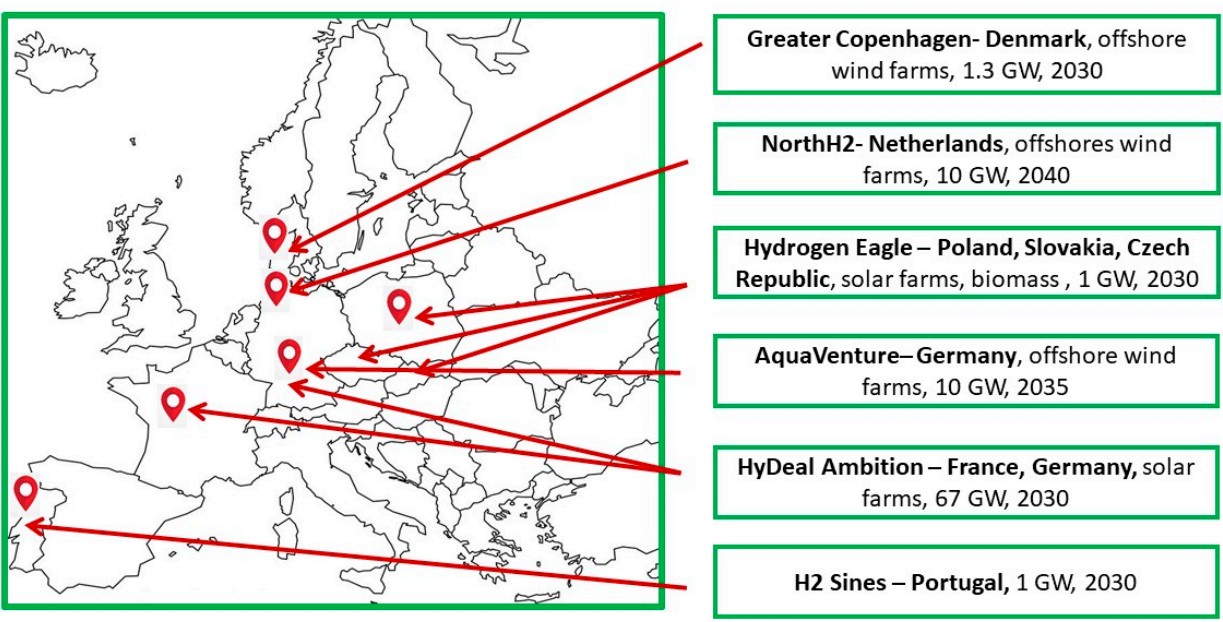

**Figure 6.** Map of hydrogen projects, own elaboration.

## 4. Development and Initiatives in the Field of Hydrogen Technologies in Poland—Road Map in Poland

Poland is a country that satisfies its energy needs mostly with energy from fossil fuels. The need to adapt to the objectives of the European Union requires the establishment of a new law that will implement these assumptions. Building the hydrogen value chain requires a legislative background. The changing energy sector must be adapted to the European energy sector in accordance with the assumptions of the European Climate Law. The construction and integration of the entire energy system based on renewable energy sources is a time-consuming and very expensive process. Moving away from coal in a country where it was the main source of energy for a long time is a big social and economic problem that will cause profound changes in the country. Over the decades of development of conventional types of energy supply from fossil fuels, an industry was created that concentrated companies specializing in this direction [53]. The changes that will take place by 2050 require a thorough analysis and transformation of this industry so that the people employed in it have a job and find themselves in the new reality. A poorly developed coal and gas phase-out plan can have dramatic economic and social consequences. At the same time, climate change and the need to reduce emissions that are harmful to health require a quick response and a profound change.

Polish plans are also very ambitious and do not differ from the EU ones. According to the provisions of the Polish Hydrogen Strategy (PHS), it provides for the greatest development in the three main sectors where hydrogen is used—transport, energy, and industry [54]. The construction of new electrolyzers (2 GW of electrolyzers are to be in operation by 2023) and the creation of a value chain (use of hydrogen in urban transport, railways, and industry) will enable economic growth by creating jobs and implementing new investments. The development of hydrogen technologies will increase the rationality of financing renewable energy sources that are part of green energy storage. The Polish Hydrogen Strategy is consistent with the European Green Deal and confirms the leading role of hydrogen, which is defined in the European Hydrogen Strategy. The Polish Hydrogen Strategy is a comprehensive, legislative hydrogen package that defines the implementation of objectives related to the elimination of barriers related to the development of the hydrogen market. In addition, it indicates detailed rules for storage, hydrogen production, technical supervision of hydrogen transport, reloading, and storage devices, health and safety rules, fire protection, and a support system for the implementation of the above-mentioned goals. The regulations in question may be included in the new comprehensive

legal act "Hydrogen law—regulatory environment supporting the development of the hydrogen economy" [55]. Technological neutrality, in accordance with the Polish Hydrogen Strategy, will determine the development of low-emission hydrogen sources (production of hydrogen from biomethane, in the process of electrolysis, from waste gases, as well as from natural gas using CCS/CSU installations) and zero-emission sources using renewable energy sources in the process of electrolysis [54].

Poland, being an EU member state, actively co-creates European law. In order to achieve the objectives of the European Green Deal and achieve climate neutrality, it is necessary to adjust national law in this direction. As of today, acts of national law are already in force in Poland, constituting the basis for the development of the hydrogen sector under the assumptions of the European Union and enabling energy transformation. On 14 February 2017, the Strategy for Responsible Development (SRD) until 2020 (with an outlook until 2030 [56]) was introduced, which is a valid and important document of the Polish state in the area of medium- and long-term economic policy. The overriding goal of the SRD is primarily to create conditions suitable for increasing the income of the inhabitants of Poland and cohesion in the economic, territorial, environmental, and social dimensions. On 7 December 2021, the full version of the "Polish Hydrogen Strategy until 2030 with a perspective until 2040" (PSW) [54] was published in Monitor Polski. As part of the strategy, it is necessary to use hydrogen technologies in the areas of industry, transport, and energy. The PSW covers a much longer period than the SRD (up to 2040), thanks to the innovative subjects of analysis. However, during detailed considerations, a period corresponding to 2030 was distinguished. The Polish Hydrogen Strategy is related to the SRD and is identified with its objectives, especially in terms of specific objective I. Sustainable economic growth is based primarily on data, knowledge, and appropriate organization. PSW uses the following SOR projects:

- Flagship project related to electromobility in Polish transport;
- Polish Nuclear Energy Program;
- Strategic Electromobility Development Program.

The draft of the Polish Hydrogen Strategy indicates that by 2030 it will be crucial to ensure appropriate conditions to start the production of hydrogen from zero- and low-emission sources. It is planned to introduce benefits that will encourage innovative activities and enable enterprises to develop and obtain financial resources. For this purpose, it is planned to use measures consisting of launching equipment for the production of low-emission hydrogen, e.g., during the electrolysis process, from waste gases, bioethane, natural gas, using CCU/CCS, and other alternative technologies for obtaining hydrogen. It is planned to use power from renewable energy sources for the production of hydrogen and synthetic fuels. The expected capacity of the installed electrolyzers is to reach 2 GW in 2030. Based on the contents of the PSW, the indicators include the production of Polish hydrogen buses in Poland in the amount of 800–1000 units and the organization of at least five hydrogen valleys [56].

PHS also takes into account coal gasification technologies, which have been recognized by the Strategy for Responsible Development (SRD) [56] as actions that may improve the country's energy security and the efficiency of the raw material in the petrochemical sector. In addition, they can contribute to an increase in efficiency in the production of materials, i.e., hydrogen and ammonia. The project of the "Polish Hydrogen Strategy" lists the six most important goals to be achieved, e.g., Figure 7 [54].

The Polish Hydrogen Strategy is also involved in the activities that are distinguished in the draft Energy Policy of Poland until 2040 (PEP 2040) [57]. PEP 2040 is a response to the important tasks facing the Polish energy sector in the near future and indicates the direction of development of the energy sector, taking into account the tasks necessary to be implemented, which include, among others, the development of the hydrogen value chain. Poland's energy policy until 2040 (PEP 2040) sets the framework for the energy transformation in Poland and is an inseparable element in the implementation of the Paris Agreement concluded in December 2015 during the XXI Conference of the Parties to the

United Nations Framework Convention on Climate Change. This document contains all determinants regarding the selection of technology used to strive for a low-emission energy system in the country. PEP2040 is one of several integrated sectoral strategies resulting from the Strategy for Responsible Development. The PEP is a supplement to the National Plan for Energy and Climate—NECP [58].

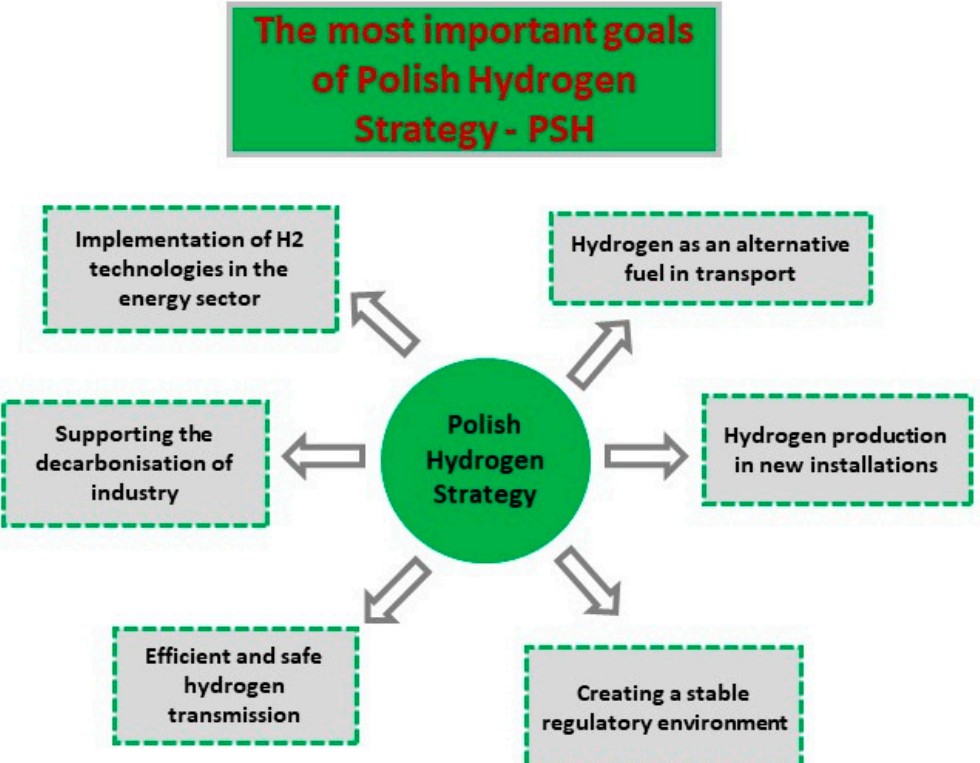

**Figure 7.** The goals of Polish Hydrogen Strategy.

Figure 8 shows the three pillars of Poland's energy policy until 2040—PEP—along with their mottos. Pillar 1 is a just transition, whose goals and assumptions are to attract new developments for regions and communities that are most vulnerable and affected by the negative effects of any activities related to the low-carbon energy transition while ensuring new jobs and building new industries. Pillar 2 concerns a zero-emission energy system, taking into account the long-term horizon through the use of nuclear energy and off-shore energy. Ensuring energy security based, for example, on gaseous fuels and thus achieving good air quality parameters are the assumptions of pillar 3. This is the pillar whose goal will be most noticeable when reducing the use of fossil fuels and, as a result, abandoning them and investing in the sectors of district heating and electromobility. Passive and zero-emission construction is also intended to improve the quality of the air, which has a significant impact on the health of society.

PEP2040 is consistent with the National Plan for Energy and Climate for 2021–2030 [58]. In 2019, Poland prepared the National Plan for Energy and Climate (NECP) (for 2021–2030). The creation of the NECP results from Regulation (EU) 2018/1999 of the European Parliament and of the Council on the Governance of the Energy Union. The purpose of the National Plan for Energy and Climate is to introduce the energy union, while the assumption of the SGP is to implement this postulate by introducing new hydrogen technologies. This document designated hydrogen as the optimal means for the development of low-emission and efficient transport. In addition, the sale of hydrogen was recognized as a good support for the profitability of the hard coal mining sector.

**Figure 8.** Pillars of Poland's energy policy until 2040 [57], own elaboration.

The Energy and Climate Policy of the European Union (EU) [59] has a very large impact on shaping and preparing the Polish energy strategy. This is a long-term plan to achieve EU climate neutrality by 2050. The implementation of these goals is an important element in achieving a low-emission energy transition. This plan assumes the decarbonization of the Member States of the European Union. In addition, in December 2020, the European Council adopted the goal of reducing net greenhouse gas emissions by 2030 by at least 55% compared to 1990 [59].

*4.1. Development of Hydrogen Technology Infrastructure and the Hydrogen Value Chain in Poland—Hydrogen Valleys*

EU law encourages the development of hydrogen technologies and the organization of hydrogen valleys through numerous subsidies for the development of hydrogen infrastructure. Due to Poland's membership in the European Union and the adopted regulations, all countries belonging to its community are obliged to adapt their laws and energy regulations to all its assumptions. Hydrogen valleys are innovative ventures that are created in places with appropriate industrial and natural resources. The hydrogen valley can be defined as an ecosystem thanks to which the so-called value chain is strictly related to the broadly understood hydrogen economy, i.e., production, transport, storage, and use of hydrogen. Within the framework of the created hydrogen valleys, in cooperation with their stakeholders, joint projects are created and implemented, and efforts are made to develop infrastructure, research, and scientific facilities. An important element on which the functioning of hydrogen valleys is based is the use of the potential of Polish enterprises for research and development of new technologies for the production and management of hydrogen.

According to the Clean Hydrogen Partnership—a public-private partnership that is the operator of EUR 2 billion of EU funds for the development of the hydrogen economy—in order to be able to talk about the hydrogen valley, five key conditions must be met, which include the following [60]:

- Scale of the project—in addition to the planned demonstration projects, stakeholders must plan at least two large, multi-million (EUR) investments as part of the valley operation. Typically, a valley consists of multiple sub-projects that make up a larger project portfolio. It is assumed that the hydrogen valley should have investment outlays of at least EUR 20 million. The European average is EUR 100 million;
- Geographically defined area—hydrogen ecosystems must cover a given area or region. It can be a local hydrogen hub and its hinterland, a region in a given country, or a cross-border region—e.g., a transport corridor along a major waterway;
- Coverage of the hydrogen value chain—i.e., activities planned in the valley from hydrogen production using energy from renewable sources through storage and distribution to its use—off-take in the region;

- The possibility of using hydrogen in several sectors of the economy—the use of regionally produced hydrogen in projects in transport, industry, and energy, applying the principle of one source—has many applications in various sectors. It is assumed that hydrogen should be used in at least two sectors of the economy;
- The activities of stakeholders in a given hydrogen valley are carried out according to a given feasibility study, which guarantees that the activities have a real chance of launching the project and obtaining financing from EU, national, and regional funds.

Based on the size of the project, three types of hydrogen valleys are distinguished in Table 2 [61].

**Table 2.** Characteristic features of particular types of hydrogen valleys.

| Types of Valleys | Descriptions |
| --- | --- |
| Type 1 | Small hydrogen valley, specialized mainly in the transport sector, for which hydrogen in the region is produced from RES in electrolyzers with a capacity of up to 10 MW, stored, and distributed for purposes such as public transport and hydrogen refueling stations, CAPEX, i.e., capital expenditures account for (production capacity), is approx. EUR 20 million. |
| Type 2 | Medium hydrogen valley, focused on decarbonization of energy-intensive industry, a type that is currently being implemented in Poland. The projects in these valleys are located around entities called anchors, i.e., large corporations and their needs. Activities within these valleys primarily integrate two sectors: industry and transport. Hydrogen is produced in electrolyzers with a capacity of up to 10–300 MW, and the capital expenditure (CAPEX) is around EUR 100 million. |
| Type 3 | A large hydrogen valley characterized by very good conditions for the production of energy and hydrogen from RES, with the use of large-scale hydrogen production for the needs of the region and for export—see: Australia (contract to Japan), Arab countries—long-term contracts, with the use of 250–1000 MW of electrolyzer capacity, and the CAPEX is approx. EUR 500 million, and more. |

*4.2. Hydrogen Valleys in Poland*

The data posted on the website of Industrial Development Agency S.A. shows that eight hydrogen valleys are currently being created in Poland, even though the Polish Hydrogen Strategy assumed the creation of five hydrogen valleys. The activity of valleys in Poland covers almost the entire area of the country; see Figure 9. The valleys that will have formed by July 2023 are the following [62]:

- Lower Silesian Hydrogen Valley;
- Mazovian Hydrogen Valley;
- Silesia—Lesser Poland Hydrogen Valley;
- Subcarpathian Hydrogen Valley;
- Central Hydrogen Cluster named after Laszczynski Brothers;
- Pomeranian Hydrogen Valley;
- Greater Poland Hydrogen Valley;
- West Pomeranian Hydrogen Valley.

Detailed data on the hydrogen valleys created in Poland are presented in Table 3. The headquarters of the hydrogen valleys are located in the largest Polish cities and bring together important companies from the energy and mining industries, which are supported by public universities.

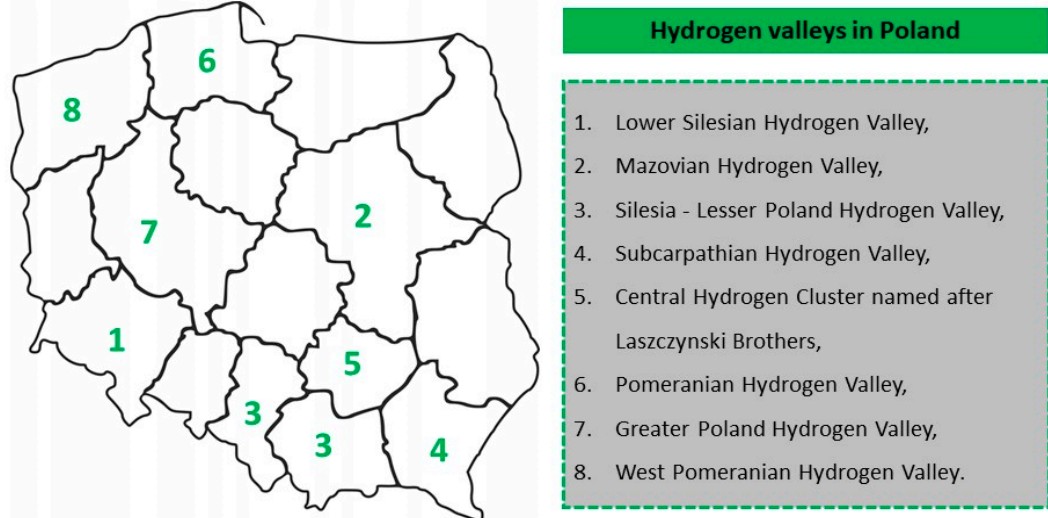

**Figure 9.** Distribution of hydrogen valleys in Poland, own elaboration.

**Table 3.** A detailed description of the hydrogen valleys formed in Poland [62].

| Valley | Headquarters | Concept | Stakeholders | Specializations | Achievements |
|---|---|---|---|---|---|
| Lower Silesian Hydrogen Valley | Wroclaw | Management of hydrogen hubs. | KGHM, ZAK Kedzierzyn-Kozle, ARP S.A. Toyota, Z-Klaster, Wroclaw University of Science and Technology, University of Wroclaw, UMWD, Linde, Total. | green ammonia, green heat, green copper and metallurgy, green river transport, hydrogen storage, RES, biogas, water pipelines. | Two applications in the consortium for Clean Hydrogen Partnership, first show at DDW in Europe—Hydrogen Week in Brussels; green $H_2$ production: 720 t/y. |
| Mazovian Hydrogen Valley | Plock | Integration to the needs of a large concern. | PKN Orlen, ARP S.A., BGK Toyota, KAPE, AGH, Siemens Energy, Warsaw University of Technology, Energy Institute University of Warsaw. | petrochemicals, synthetic fuels, production of green hydrogen, biogas, HRS, hydrogen logistics, energy storage, green chemistry. | Petrochemicals, synthetic fuels, production of green hydrogen, biogas, HRS, hydrogen logistics, energy storage, green chemistry. |
| Silesia—Lesser Poland Hydrogen Valley | Katowice | Energy transformation in Silesia and Lesser Poland based on FSI. | Orlen South, Polenergia, ARP S.A., JSW Innovation, Azoty Group, The Silesian Technical University, AGH, KOMAG, IPTE, Katowicka SSE, GZM, Columbus. | green glycol, energy transformation of Silesia, hydrogen production, hydrogen storage, green steel, zero-emission public transport, CCUS. | Commissioning of the installation for the production of green glycol; H2Poland portal. |
| Subcarpathian Hydrogen Valley | Rzeszow | Integration around the university and the aviation valley. | Rzeszów University of Technology, Podkarpackie Marshal's Office, Pole-nergia, city of Sanok, ARP S.A., entities from the aviation valley, Autosan, ML System. | hydrogen buses, aviation, hydrogen in the energy sector, green heat, hydrogen-strings. | Application for project development assistance. |
| Central Hydrogen Cluster named after Laszczynski Brothers | Kielce | Integration around the project of decarbonisation of raw material mines. | Industrial Group Industria, ARP S.A., city of Kielce, commune of Chęciny, Świętokrzyska University of Technology, Colum-bus, ML System, AIUT, Azoty Group. | hydrogen production, hydrogen dump trucks, hydrogen storage, RES, green public transport. | Feasibility study; transition to the implementation phase. |

**Table 3.** *Cont.*

| Valley | Headquarters | Concept | Stakeholders | Specializations | Achievements |
|---|---|---|---|---|---|
| Pomeranian Hydrogen Valley | Gdansk | Integration around local government initiatives, decarbonisation of the port of Gdynia, concept of "Shore H2 Valley". | Pomeranian Marshal's Office, Cluster of Hydrogen Technologies, city of Gdynia, PKP Energetyka, port of Gdynia, Sescom, Gdańsk University of Technology. | zero-emission public transport, hydrogen production, offshore, port decarbonisation, hydrogen storage, production of electrolyzers, HRS. | Completed mapping of the region's potential in PDA format. |
| Greater Poland Hydrogen Valley | Poznan | Integration around local government initiatives and the needs of a large corporation. | Wielkopolska Marshal's Office, ZE PAK, Solaris, Adam Mickiewicz University, Poznań University of Technology, ARR Konin, city of Piła, Wielkopolska Council of Thirty | hydrogen production, hydrogen storage, HRS, hydrogen buses, RES. | Mapping the needs of the region; h2wielkopolska.pl portal. |
| West Pomeranian Hydrogen Valley | Szczecin | Integration around a large chemical concern; the "Shore H2 Valley" concept. | West Pomeranian University, Azoty Group, ARP S.A., Enea, Port Police, Maritime University of Technology, NFOŚ, ZUT, Koszalin University of Technology. | green ammonia, low-emission sea transport, low-emission river transport, ammonia collection infrastructure, hydrogen production, offshore. | Over 30 entities interested in the development of the hydrogen valley. |

Hydrogen transport, especially urban transport, is of great importance in the development of hydrogen technologies. Several investments related to the construction of hydrogen stations are planned in Poland. The first places where it will be possible to refuel hydrogen are Warsaw and Gdańsk. Stations such as Lotos in Gdańsk (Benzynowa Street) and in Warsaw (Łopuszańska Street) will also be equipped with a hydrogen refueling dispenser. Hydrogen H70 (700 bar) and H35 (350 bar) will be available at these locations. Hydrogen dispensers will specify the amount of fuel in kilograms. The available map of hydrogen stations is supplemented on the basis of information from hydrogen suppliers and information on hydrogen fuel. The Ministry of Climate and Environment submitted a draft act amending the act on electromobility and alternative fuels and other related acts to the Committee for European Affairs for consideration. Therefore, the phase of inter-ministerial arrangements and social consultations was completed. The most important changes in the project include, among others:

- A proposal for a new wording of the regulations regarding the area of clean transport, which are mandatory for cities with more than 100,000 inhabitants. They show an average annual exceedance of nitrogen dioxide ($NO_2$) pollution. A modification of the catalog of vehicles authorized to enter the zone was also included. In addition, it has been possible for municipal authorities to assign separate exemptions related to vehicle traffic;
- Reducing the amount and frequency of collecting fees related to EIPA numbers in order to significantly accelerate the expansion of the vehicle charging network;
- Changing the management board's obligations regarding the installation and use of charging points in buildings to the designated power level and shortening the period for examining applications;
- Introduction of the concept of an electrically assisted bicycle, which ensures the development and assistance for the most diverse electric vehicles, and improvement of definitions related to the use of hydrogen in transport.

The Electromobility and Alternative Fuels Act of 11 January 2018 set out the rules for the development and operation of infrastructure for the use of alternative fuels in transport,

referred to as "alternative fuels infrastructure", including the technical requirements to be met by this infrastructure; obligations of public entities in the development of alternative fuels infrastructure; information obligations in the field of alternative fuels; conditions for the functioning of clean transport zones; a national policy framework for the development of alternative fuels infrastructure; and the manner of their implementation. Figure 10 presents plans for the expansion of hydrogen hubs along with the cities where they will be located, as well as the number of hydrogen refueling stations in Poland, the Czech Republic, and Slovakia, as well as hydrogen production sources in the PKN ORLEN S.A. group—a Polish energy giant. The largest hydrogen hub in the ORLEN Group will be the hub in Wloclawek, whose target hydrogen production is 600 kg/h. At the initial stage of distribution, fuel from the hub in Wloclawek will be used for public transport and later also for freight transport. A similar installation of this type will also be built in Wloclawek and in the ORLEN South biorefinery in Trzebinia [50]. The map includes the existing hydrogen production plants: Hub Plock, Wloclawek, and Trzebinia. Production of hydrogen from RES from off-shore and on-shore: Hub North, Hub Bystra, Hub South, and the possibility of processing municipal waste into hydrogen: Hub Plock and Hub Ostroleka. PKN ORLEN S.A. constantly invests in modern and environmentally friendly hydrogen technologies. As part of "Clean Cities—Hydrogen mobility in Poland (Phase I)" [63], one of the largest national projects in terms of hydrogen production volume, the concern will build two public hydrogen refueling stations in Poznan and Katowice and a mobile station in Wloclawek. For the implementation of this project, PKN ORLEN received EUR 2 million in funding from the EU CEF Transport Blending Facility program. By 2030, the company plans to build over 50 hydrogen refueling stations throughout Poland [64].

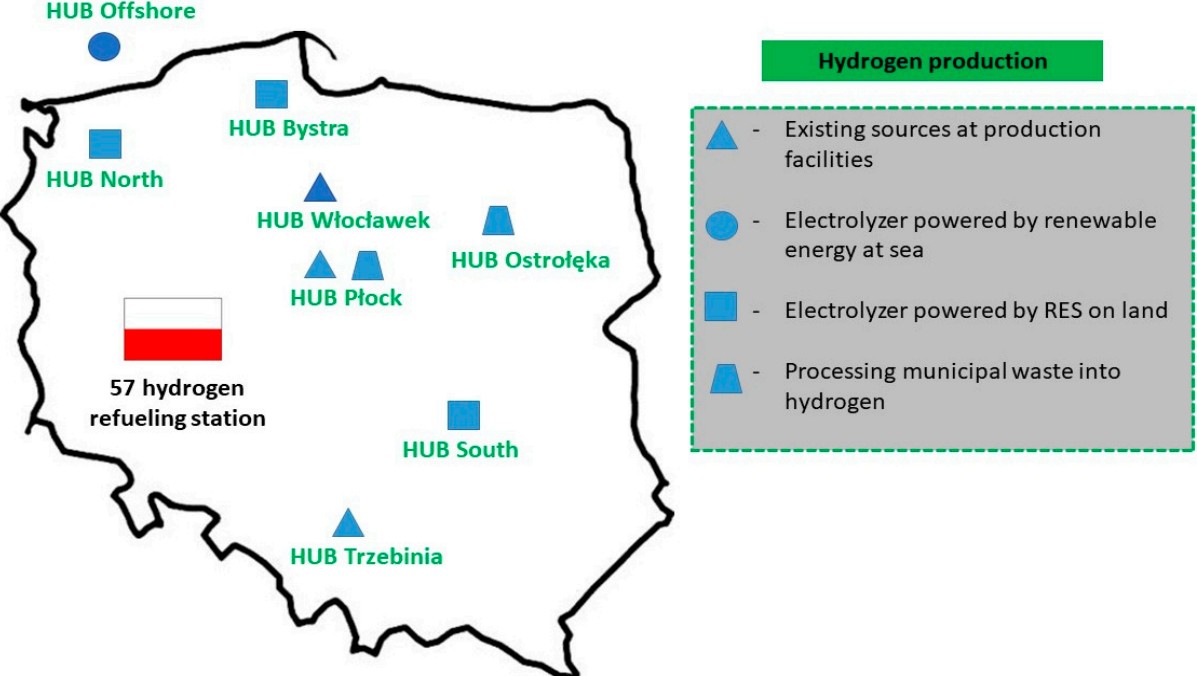

**Figure 10.** Plans to expand hydrogen hubs, refueling stations, and hydrogen production plants in the ORLEN Group, own elaboration.

*4.3. Analysis of Hydrogen Production and Storage Possibilities in Polish Conditions*

Poland is ranked fifth in the global ranking of hydrogen producers. Despite this, the share of electrolyzers is imperceptible. Annual hydrogen production is approximately 8.8 billion $m^3$. This hydrogen is mainly produced by chemical plants and refineries and used in the production and refining of artificial fertilizers. The companies involved in the development of the hydrogen economy include primarily state-owned companies. In July

2020, a letter of intent on the development of the hydrogen economy was signed by a total of eighteen entities. Eight of them are state-owned companies (subordinated to the MAP), one university, three state-owned research institutes, two industry organizations, and three private enterprises. In the hydrogen value chain, hydrogen production is an important aspect. There are currently four main hydrogen producers in Poland:

- Azoty Group—42% market share (producing approx. 420,000 tons/year, a significant part of the hydrogen is sold on the market—covering approx. 85% of domestic sales);
- LOTOS Group—14% market share (producing approx. 145,000 tons/year, used for own purposes);
- PKN Orlen—14% share (producing approx. 145,000 tons/year, used for own purposes);
- Jastrzebska Spolka Weglowa—7% share (producing approx. 75 thousand tons per year);
- Others: 23%.

Thanks to the invention of Lotos Group, in 2018 the Cluster of Hydrogen Technologies and Clean Energy Technologies was established, the main goal of which is to increase the value of hydrogen technologies. The company concluded an agreement with the authorities of the cities of Wejherowo and Gdynia for the supply of hydrogen intended for driving buses. There is a high probability that Lotos Group will be the first producer of purified hydrogen in Poland, which can be used in transport based on fuel cells. In addition, it is possible to build the first stations that will enable refueling with pure hydrogen. In 2018, during the Conference of the Parties to the United Nations Framework Convention on Climate Change (COP24 [65]) in Katowice, an agreement was concluded for the reimbursement of the "Pure H2" project [66,67], the assumption of which was the construction of an installation for the recovery of hydrogen in refineries and a suitable refueling station for this fuel for vehicles. The price of the investment in question is approximately EUR 10 million.

Lotos is also introducing projects related to the construction of prototype shunting locomotives with hybrid drives and has started cooperating with the academic side. The company signed an agreement with a company planning to develop offshore wind farms in the Baltic Sea. The main assumption was the implementation of the initial hydrogen production project using the water electrolysis process.

Other companies implementing development projects in the area of hydrogen include, among others, PGNiG, JSW Group, and Energa. The JSW Group is working on adapting the hydrogen separation solution from coke oven gas, thanks to which it is possible to obtain a high-purity product. Hydrogen obtained in this way complies with the purity requirements and is dedicated primarily to electromobility. The company was the first in Poland to announce joining the European Association of Hydrogen and Fuel Cells— "Hydrogen Europe" [68]. The association in question also includes global companies that are the most advanced in the development of hydrogen-related technologies. These include, e.g., Airbus, 3M, Anglo American, Alstom, BMW Group, Audi, China Energy, Toyota, and General Motors.

In 2018, PGNiG launched the "ELIZA" [69] research project, which focuses primarily on the analysis of the possibility of producing hydrogen from renewable energy sources by electrolysis and the technology of injecting it into storage facilities used to store natural gas. The production of green hydrogen has great potential in Poland, which results from the number of installed photovoltaic (PV) panels thanks to EU funding. As of February, the number of installed PVs for 2023 was 1,193,051. Only in December last year, 14,245 new PV installations with a capacity of 334.56 MW were installed, which makes it possible to connect electrolyzers and produce green hydrogen from energy surpluses. The installed PV power is, in a sense, translated into the number of fuel cell units. When analyzing the production of green hydrogen on an industrial scale with the use of photovoltaic farms, it was planned that 1–2 large installations (on a scale of several MW) would be built in the near future, as declared by Polish institutions [43,52].

Table 4 presents Polish companies operating in the hydrogen sector along with the areas of their activity supporting the development of hydrogen technologies. Poland

has leaders in every link of the hydrogen value chain: production, storage, use, and transmission, along with access to specialized equipment. Poland is the third producer of hydrogen, but in order to meet the assumptions of the Green Deal, this hydrogen should have low or zero emissions. Domestic companies clearly emphasize that the hydrogen market in Poland is currently at an early stage. Entities that can benefit the most from the future hydrogen revolution already see the potential of new technologies. The size of the hydrogen market in Poland can be estimated at approx. 752,000 tons per year. On the other hand, the total volume of the hydrogen market in Poland can be calculated at about 1.6 million tons per year. Most of this volume falls in the hydrogen purity class in the range of 2.5–4.0 (99.5–99.99%), used by refineries, chemical plants, energy, and metallurgy. The highest purity hydrogen (class 5.0–99.999%) is used in the food, chemical, and electronic industries and laboratories. Hydrogen belonging to class 5.0 is imported in the amount of 226 tons per year. Polish organizations believe that the domestic industry has a chance to convert "grey" hydrogen (produced in the process of reforming natural gas or other hydrocarbons produced in the process of oil refining) to "blue" (produced in processes using fossil fuels, supplemented with technologies for capturing, storing, or processing $CO_2$), but this should not only apply to production from gas. The condition for including hydrogen in the low-emission mode is the avoidance of $CO_2$ emissions, storage (CCS), and reuse of this gas (CCU). Waste hydrogen, which is a by-product of ammonia synthesis (purge gas), can also be mentioned. When producing propene from propane, waste hydrogen is also produced without $CO_2$ emissions.

An important task is the possibility of replacing gray hydrogen, which comes from gas, with green hydrogen, which is produced using energy and is widely practiced in Germany. The support system for green hydrogen and the possibilities of its use in refineries present broad perspectives. Solutions of this type are treated as a way to reduce $CO_2$ emissions, e.g., in air transport, and as the beginning of creating market space for P2G solutions. An important factor related to the development of RES is repeatedly emphasized, which will significantly determine the development of the hydrogen sector. The report of FCH JU "Hydrogen Roadmap Europe" [70] shows that the level of production of class 2.5 to 4.0 hydrogen for industrial purposes will remain at the current level. It can be assumed that the importance of hydrogen with purity class 5.0, which is intended for fuel cells, will increase significantly.

An undoubted advantage of using hydrogen as an alternative to fossil fuels is the fact that it can be produced from renewable sources that do not emit $CO_2$, and its application is much wider than before. Most of the hydrogen produced today comes from fossil fuels. With the development of RES and the falling costs of green energy production, the production of hydrogen in electrolyzers becomes possible.

It is estimated that investments related to the production of emission-free hydrogen will reach EUR 470 billion by 2050 [70,71]. The tasks set for Poland, a large European producer of gray hydrogen, must be considered. In order to determine what the European Hydrogen Strategy entails for Poland, it is necessary to precisely define the term "Clean hydrogen", and, in particular, what value of $CO_2$ emissions will qualify for recognition as producing low-emission hydrogen. Such an agreement is necessary at the European level, taking into account the just transition of $CO_2$-intensive regions. In developing a common low-emission standard, the values developed should reflect our point of view but, on the other hand, be consistent with the overriding goal of the EU, i.e., decarbonization of the economy. It may be necessary to prepare a justification based on technical and economic arguments that would support the modernization of existing hydrogen-generating installations in Poland, taking into account their emission intensity throughout the entire life cycle [72].

**Table 4.** Polish companies with experience in selected segments of the hydrogen economy, own elaboration.

| Hydrogen Value Chain | Activity Area | Stakeholder |
| --- | --- | --- |
| Production | separation of hydrogen from coke oven gas | Jastrzębska Społka Weglowa (JSW) |
| | use of electrolyzers to produce H2 from renewable energy sources | Lotos, Polish Power Grids (PSE) |
| | sale of electrolyzers powered by PV | Sescom |
| | scaling own production of "grey" hydrogen for sale | Azoty Group |
| | production and use of "green" hydrogen; cogeneration converted to hydrogen combustion | Polenergia |
| | distribution of electrolyzers | RB Consulting |
| | the use of electrolyzers with electricity from biomass | Pątnów Adamów Konin Power Plant Complex (ZE PAK) |
| | the use of electrolyzers in cooperation with renewable energy sources | Orlen |
| | production of SNG (synthetic natural gas): hydrogen from electrolysis with electricity from RES; carbon dioxide from emission installations | Tauron Wytwarzanie |
| | separation of hydrogen from coke oven gas | Walbrzych Plants Coke "Victoria" |
| | methane steam reforming | Steelproduct |
| Storage | injection of hydrogen into distribution and transmission networks; underground storage | PGNiG |
| | Stako from the Worthington Industries Group | tanks |
| Application | vehicle charging stations | Lotos, PKN ORLEN, PGNiG |
| | use of hydrogen in locomotives | PKP Cargo |
| | hydrogen locomotive prototype | H. Cegielski |
| | development of fuel cell models | EC Grupa (Energocontrol Sp. z o.o.) |
| | hydrogen emergency power system | APS Energia with the Gdańsk University of Technology |
| | hydrogen bus production | Solaris |
| | combustion of hydrogen in cogeneration turbines | Polenergia (z Siemens) |
| Transmission in gas networks | hydrogen blending in gas pipelines | PGNiG |
| Apparatus and devices | gas meters | Intergas |
| | pressure tanks | cGAS controls |
| | hydrogen sensors | Emag Serwis |
| | cryostatic devices | Frankoterm |

## 5. Summary

Poland's role in the face of the tasks set by the European Union is to adapt its energy economy so that it is as low-emission as possible. In pursuit of this goal, it will be necessary to develop technologies for the production and use of hydrogen due to the wide range of hydrogen applications, from the role of a raw material through fuel to energy storage. In implementing the European goals related to the reduction of $CO_2$ emissions, it should be emphasized that the technologies that will be supported are technologies for the production of green hydrogen and gray hydrogen. Trends related to the management of energy surpluses from RES are also clearly visible in Poland. As part of the assumptions of the European Hydrogen Strategy, in the years 2020–2024, it is planned that electrolyzers with a capacity of at least 6 GW, powered by renewable energy, will be installed. Therefore, it is assumed that the production of electrolyzers will increase, in particular those with the highest capacity—up to 100 MW—which can be placed next to existing hydrogen demand centers in steelworks, refineries, or chemical complexes. It is also assumed that in the next phase, planned for 2025–2030, hydrogen will be part of the integrated energy system. The capacity installed in electrolyzers can reach up to 40 GW. Hydrogen production based on RES can guarantee predictability and flexibility for renewable energy sources. It is

estimated that by the end of 2030, pure/green hydrogen will be priced competitively with grey/conventional hydrogen. In the period between 2030 and 2050, technologies for the production of clean hydrogen should be so cost-balanced and advanced that their use should be introduced on a large scale, which will contribute to the actual decarbonization of the economy.

Poland does not lag behind in reducing $CO_2$ emissions and decarbonizing the economy. Poland, as a member of the EU, is the creator of documents that take into account the assumptions of the European Union Commission and systematically implement the assumed goals. Regardless of the country's wealth, the key aspect is the co-financing of projects supporting processes for the development of infrastructure for hydrogen transport (transport via pipelines and off-grid transport—road, rail, etc.) and, first of all, the reconstruction of the already existing gas infrastructure. The hydrogen market forces the construction of a large-scale infrastructure for its storage and an appropriate network of refueling stations; therefore, appropriate actions have already been taken in Poland. The strategy of activities supporting the development of hydrogen technologies in Poland and the value chain includes the following processes:

- Implementation of assumptions in accordance with the recommendations of the European Commission regarding the need to implement zero-emission methods of obtaining hydrogen;
- Implementation of the objectives and assumptions of the Polish Hydrogen Strategy and supporting documents, including, in particular, the Energy Policy of Poland PEP2040, based on a just transformation, a zero-emission energy system, and the pursuit of good air quality;
- Development of hydrogen technologies in the three main sectors where hydrogen is used—transport, energy, and industry;
- Striving for economic growth by creating jobs and investments in the creation of the hydrogen value chain, including the construction of new electrolyzers,
- Development of hydrogen technologies in the field of initiatives in the form of clusters and hydrogen valleys;
- Activities of hydrogen valley stakeholders and Polish leaders in the energy industry, taking into account long-term plans supported by programs subsidizing activities related to hydrogen technologies;
- International cooperation between Poland, Slovakia, and the Czech Republic under the project HydrogenEagle, whose main assumption is the production of green hydrogen using biomass and solar farms in electrolyzers with a capacity of 1 GW;
- Activities related to gathering specialized staff in the field of hydrogen technologies by creating didactic courses, organizing training, and post-graduate studies;
- Plans to create 57 hydrogen refueling stations in Poland, obtained in hubs from four sources;
- Poland is ranked fifth in the global ranking of hydrogen producers and third in hydrogen production in Europe;
- Implementation of projects based on the production of hydrogen from biomethane and the processing of municipal waste, which is also tantamount to recycling activities.

**Author Contributions:** Conceptualization, R.W.; methodology, P.K.; investigation, P.K.; writing—original draft preparation, R.W.; writing—review and editing, R.W. All authors have read and agreed to the published version of the manuscript.

**Funding:** The scientific research was funded by the statute subvention of Czestochowa University of Technology, Faculty of Infrastructure and Environment. The research was funded by project No. BS/PB400/301/23.

**Institutional Review Board Statement:** Not applicable.

**Informed Consent Statement:** Not applicable.

**Data Availability Statement:** Not applicable.

**Conflicts of Interest:** The authors declare no conflict of interest. The funders had no role in the design of the study; in the collection, analyses, or interpretation of data; in the writing of the manuscript, or in the decision to publish the results.

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
