# Peer review of "Modern Hydrogen Technologies in the Face of Climate Change—Analysis of Strategy and Development in Polish Conditions"

_sustainability, doi:10.3390/su151712891_

Round 1
Reviewer 1 Report
- Line 19: “As noted in this article, Poland is one of the leaders in hydrogen production both in the world and in Europe.” Please eliminate the “As noted in this article”.
- The results of the research should be given in the abstract. Please state the results clearly.
- Keywords should be sorted alphabetically.
- Line 30: The sentence “The answer to problems related to the deteriorating quality of the natural environment, the depletion of fossil resources and the ever-increasing demand for electricity is hydrogen as a modern energy carrier.” should be rewritten and checked for grammar.
- Line 74: the sentence “The hydrogen produced in the electrolyser is directly sent to the fuel cell or stored. In this way, fuel cells can replace photovoltaic panels during high cloud cover or at night.” needs a references and It is also better to mention some previous researches in this field.
- One of the weaknesses of this section is the lack of literature reviews. Authors should focus more on past research.
- Line 252: Please correct “CO2”.
Author Response
Reviewer #1
Answer to Reviewer #1
Thank you very much for all your comments and time. Below we present the changes that were made to the manuscript and the responses to the remarks of the honorable Reviewer:
- Line 19: “As noted in this article, Poland is one of the leaders in hydrogen production both in the world and in Europe.” Please eliminate the “As noted in this article”.
Modified:
As noted in this article, Poland is one of the leaders in hydrogen production both in the world and in Europe.
- The results of the research should be given in the abstract. Please state the results clearly.
Completed in Abstract chapter:
The review of the analysis of the development of hydrogen technologies in Poland shows that Poland does not differ from other European countries. As part of the assumptions of the European Hydrogen Strategy and the trend related to the management of energy surpluses, electrolyzers with a capacity of at least 6 GW will be installed in Poland in 2020-2024. It is also assumed that in the next phase, planned for 2025-2030, hydrogen will be a carrier in the energy system in Poland. Poland, as a member of the EU, is the creator of documents that take into account the assumptions of the European Union Commission and systematically implements the assumed goals. The strategy of activities supporting the development of hydrogen technologies in Poland and the value chain includes very extensive activities related to, among others, obtaining hydrogen, using hydrogen in transport, energy and industry, developing human resources for the new economy, supporting the activities of hydrogen valley stakeholders, building hydrogen refueling stations and cooperation Poland with Slovakia and the Czech Republic as part of the HydrogenEagle project.
- Keywords should be sorted alphabetically.
Modified:
green hydrogen, hydrogen economy, hydrogen technology, hydrogen valleys, Polish Hydrogen Strategy
- Line 30: The sentence “The answer to problems related to the deteriorating quality of the natural environment, the depletion of fossil resources and the ever-increasing demand for electricity is hydrogen as a modern energy carrier.” should be rewritten and checked for grammar.
Modified:
Hydrogen is a modern energy carrier. Technologies that allow the use of hydrogen in the short term will be a solution to the problems of the deteriorating quality of the natural environment, the depletion of fossil resources and the ever-growing demand for electricity.
- Line 74: the sentence “The hydrogen produced in the electrolyser is directly sent to the fuel cell or stored. In this way, fuel cells can replace photovoltaic panels during high cloud cover or at night.” needs a references and It is also better to mention some previous researches in this field.
Completed and modified:
The hydrogen produced in the electrolyser is directly sent to the fuel cell or stored [5-7].
- Final Report: Assessment of Electrolysers, Scottish Government, Riaghaltas na h-Alba, October 2022
- Shan Wang, Aolin Lu, Chuan-Jian Zhong, Hydrogen production form water electrolysis: role of catalysts, Nano Convergence, 8, 4, 2021, https://doi.org/10.1186/s40580-021-00254-x
- Alexander Buttler, Hartmut Spliethoff, Current status of water electrolysis for energy storage, grid balancing and sector coupling via power-to gas and power-to-liqiuds: A review, Renewable and Sustainable Energy Reviews, 82, (2018, 2440-2452, http://dx.doi.org/10.1016/j.rser.2017.09.003
- Youssef Naimi, Amal Antar, Hydrogen generation by water electrolysis, http://dx.doi.org/10.5772/intechopen.76814
- One of the weaknesses of this section is the lack of literature reviews. Authors should focus more on past research.
Completed additional literature reviews:
[11] T. Tani, N. Sekiguchi, M. Sakai, D. Ohta, Optimization of solar hydrogen systems based on hydrogen production cost, Solar Energy, 68 (2), 2000, 143-149, https://doi.org/10.1016/S0038-092X(99)00061-4
[13] Haris Ishaq, Ibrahim Dincer, Curran Crawford, A review on hydrogen production and utilization: Challenges and opportunities, Int. j. Hydrogen Energy, 47 (62), 2022, 26238-26264, https://doi.org/10.1016/j.ijhydene.2021.11.149
[14] Cost- competitive green hydrogen: how to lower the cost of electrolysers? The Oxford Institute for Energy Studies, January 2022, ISBN 978-1-78467-193-8
[15] Guandalini, G., Campanari, S., and Valenti, G. (2016) ‘Comparative assessment and safety issues in state-of-the-art hydrogen production technologies’, International Journal of Hydrogen Energy, 41 (42), pp. 18901–20
[21] Meiling Yue, Hugo Lambert, Elodie Pahon, Robin Roche, Samir Jemei, Daniel Hissel, Hydrogen energy systems: A critical review of technologies, applications, trends and challenges, Renewable and Sustainable Energy Reviews, 146, 2021, 111180, https://doi.org/10.1016/j.rser.2021.111180
[22] Sakshee Rai, Mahima Habil, David Massey, A review on green hydrogen: An alternative of climate change mitigation, Indian Journal of Advanced in Chemical Science, 9(4), 2021, 340-345.
[23] The Future of Hydrogen, Seizing today’s opportunities, Chapter 2: Producing hydrogen and hydrogen-based production, Report prepared by the IEA for the G20, Japan 2019.
[37] www.weforum.org, What is green hydrogen? An expert explains its benefits | World Economic Forum (weforum.org) (accessed on August 2023)
[38] https://www.europarl.europa.eu/ Renewable hydrogen: what are the benefits for the EU?
(accessed on August 2023)
[71] www.intereconomics.eu, Green Hydrogen in Europe: Do Strategies Meet Expectations? (accessed on August 2023)
- Line 252: Please correct “CO2”.
Modified:
This is a significant increase that will contribute to decarbonisation and improve air quality and reduce CO2 emissions.
Reviewer 2 Report
The manuscript, "Modern Hydrogen Technologies in the Face of Climate Change - Analysis of strategy and development in Polish Conditions" is an exciting topic and summarizes the hydrogen-related activities of Poland. I would suggest the following points to make the manuscript easier for the readers to understand.
1. The way the manuscript is structured is difficult to understand. I suggest writing a paragraph or a flowchart to express the organization of the manuscript.
2. The topics such as the generation of electrolyzers, 3 phases of the EU hydrogen strategy shall be written in the form of charts to make the readers grasp the content in minimum time possible.
3. The green hydrogen and the electricity generation (also demand & surplus) always go together. So, few topics inclusion about electricity generation and the future outlook of electricity generation of Poland in short is recommended.
4. As the entire manuscript is about the road map of hydrogen technologies, the novelty comes from the fact of inclusion of charts/tables/infographics to make the readers easier to understand in the minimum time possible rather than reading bunches of sentences. Hence, wherever possible, rewrite the long sentences into charts/tables and infographics.
Author Response
Reviewer #2
Answer to Reviewer #2
Thank you very much for all your comments and time. Below we present the changes that were made to the manuscript and the responses to the remarks of the honorable Reviewer
The manuscript, "Modern Hydrogen Technologies in the Face of Climate Change - Analysis of strategy and development in Polish Conditions" is an exciting topic and summarizes the hydrogen-related activities of Poland. I would suggest the following points to make the manuscript easier for the readers to understand:
- The way the manuscript is structured is difficult to understand. I suggest writing a paragraph or a flowchart to express the organization of the manuscript.
Completed and modified:
In order to present the activities that are already taking place in the development of hydrogen technologies in Poland and in the world, it was necessary to refer to the legal regulations of the European Union and Poland. These documents set the overarching goals of reducing pollution, searching for new energy carriers, and developing innovations. Figure 1 shows the organization of the manuscript.
Fig. 1. Manuscript organization diagram. The graph in the attached file.
- The topics such as the generation of electrolyzers, 3 phases of the EU hydrogen strategy shall be written in the form of charts to make the readers grasp the content in minimum time possible.
Completed and modified:
Fig. 3. Development of technology for electrochemical production of hydrogen in electrolysers – generations of electrolyzers. The graph in the attached file.
Fig. 5. Phases of the EU Hydrogen Strategy for 2020-2050 The graph in the attached file.
- The green hydrogen and the electricity generation (also demand & surplus) always go together. So, few topics inclusion about electricity generation and the future outlook of electricity generation of Poland in short is recommended.
Completed and modified:
In 2018, PGNiG launched the “ELIZA” [69] research project, which focuses primarily on the analysis of the possibility of producing hydrogen from renewable energy sources by electrolysis and the technology of injecting it into storage facilities used to store natural gas.
The production of green hydrogen has great potential in Poland, which results from the number of installed photovoltaic (PV) panels, thanks to EU funding. As of February, the number of installed PV for 2023 was 1,193,051. Only in December last year, 14,245 new PV installations with a capacity of 334.56 MW were installed, which makes it possible to connect electrolysers and produce green hydrogen from energy surpluses. The installed PV power is in a sense translated into the number of fuel cell units. When analyzing the production of green hydrogen on an industrial scale with the use of photovoltaic farms, it was planned that 1-2 large installations (on a scale of several MW) will be built in the near future, as declared by Polish institutions [43, 52].
It is estimated that investments related to the production of emission-free hydrogen will reach EUR 470 billion by 2050 [70-71]. Considering the tasks set for Poland, a large European producer of gray hydrogen. In order to determine what the European Hydrogen Strategy entails for Poland, it is necessary to precisely define the term "Clean hydrogen", and in particular what value of CO2 emissions will qualify for recognition as producing low-emission hydrogen. Such an agreement is necessary at European level, taking into account the just transition of CO2-intensive regions. Developing a common low-emission standard. The values developed should reflect our point of view, but on the other hand, be consistent with the overriding goal of the EU, i.e. decarbonisation of the economy. It may be necessary to prepare a justification based on technical and economic arguments that would support the modernization of existing hydrogen-generating installations in Poland, taking into account their emission intensity throughout the entire life cycle [72].
- As the entire manuscript is about the road map of hydrogen technologies, the novelty comes from the fact of inclusion of charts/tables/infographics to make the readers easier to understand in the minimum time possible rather than reading bunches of sentences. Hence, wherever possible, rewrite the long sentences into charts/tables and infographics.
Completed and modified:
The project of the "Polish Hydrogen Strategy" lists the 6 most important goals to be achieved, e.g. [41]:
- Objective 1 - "implementation of hydrogen technologies in the energy sector";
- Objective 2 – "use of hydrogen as an alternative fuel in transport";
- Objective 3 - "supporting the decarbonisation of industry";
- Objective 4 - "production of hydrogen in new installations";
- Objective 5 – "efficient and safe hydrogen transmission";
- Objective 6 – “creating a stable regulatory environment”.
The graph in the attached file.
Fig. 7. The goals of Polish Hydrogen Strategy
Based on the size of the project, three types of hydrogen valleys are distinguished [48]:
- Type 1 - small hydrogen valley, specialized mainly in the transport sector, for which hydrogen in the region is produced from RES in electrolysers with a capacity of up to 10 MW, stored and distributed for purposes such as public transport and hydrogen refueling stations, CAPEX, i.e. capital expenditures account for (production capacity) is approx. EUR 20 million.
- Type 2 - medium hydrogen valley, focused on decarbonisation of energy-intensive industry, a type that is currently being implemented in Poland. The projects in these valleys are located around entities called anchors, i.e. large corporations and their needs, activities within these valleys primarily integrate two sectors: industry and transport. Hydrogen is produced in electrolysers with a capacity of up to 10-300 MW, and the capital expenditure (CAPEX) is around EUR 100 million.
- Type 3 - a large hydrogen valley characterized by very good conditions for the production of energy and hydrogen from RES, with the use of large-scale hydrogen production for the needs of the region and for export - see: Australia (contract to Japan), Arab countries - long-term contracts, with the use of 250-1000 MW of electrolyzer capacity, CAPEX of approx. EUR 500 million and more
The Table in the attached file.
Table 2. Characteristic features of particular types of hydrogen valleys
|
Types of valleys |
Descriptions |
|
Type 1 |
small hydrogen valley, specialized mainly in the transport sector, for which hydrogen in the region is produced from RES in electrolysers with a capacity of up to 10 MW, stored and distributed for purposes such as public transport and hydrogen refueling stations, CAPEX, i.e. capital expenditures account for (production capacity) is approx. EUR 20 million. |
|
Type 2 |
medium hydrogen valley, focused on decarbonisation of energy-intensive industry, a type that is currently being implemented in Poland. The projects in these valleys are located around entities called anchors, i.e. large corporations and their needs, activities within these valleys primarily integrate two sectors: industry and transport. Hydrogen is produced in electrolysers with a capacity of up to 10-300 MW, and the capital expenditure (CAPEX) is around EUR 100 million. |
|
Type 3 |
a large hydrogen valley characterized by very good conditions for the production of energy and hydrogen from RES, with the use of large-scale hydrogen production for the needs of the region and for export - see: Australia (contract to Japan), Arab countries - long-term contracts, with the use of 250-1000 MW of electrolyzer capacity, CAPEX of approx. EUR 500 million and more. |
Round 2
Reviewer 1 Report
I wish the authors success in their future research